# Study on Staged Damage Behaviors of Rock-like Materials with Different Brittleness Degrees Based on Multiple Parameters

**DOI:** 10.3390/ma16062334

**Published:** 2023-03-14

**Authors:** Tong Jiang, Li Wan, Wenxue Wang, Chao Xu, Chen Liu, Fanke Meng, Yuan Cui, Longfei Li

**Affiliations:** College of Geosciences and Engineering, North China University of Water Resources and Electric Power, Zhengzhou 450046, China

**Keywords:** multiple parameters, brittleness degree, instability precursor information, natural frequency, uniaxial compression

## Abstract

Understanding the brittle fracture behavior of rock is crucial for engineering and Earth science. In this paper, based on acoustic emission (AE) and laser Doppler vibration (LDV) monitoring technology, the staged damage behaviors of rock-like materials with different brittleness degrees under uniaxial compression are studied via multiple parameters. The results show that the brittleness degree determines the fracture mode. As the specimen’s brittleness degree increases, the tensile failure increases and shear failure decreases. AE activity is enhanced at the crack damage point. With an increasing specimen brittleness degree, different instability precursor information is shown during the unstable crack growth stage: the AE *b* value changes from the fluctuating to continuously decreasing state, and the natural frequency changes from the stable fluctuation to upward fluctuation state. The AE *b* value near the stress drop is the smallest, and it decreases with an increasing brittleness degree. The natural frequency reduction indicates the rock-like fracture. The natural frequency is a symbolic index that reflects staged damage characteristics and predicts the amount of energy released by brittle failure. These findings provide guidelines for rock stability monitoring and provide support for better responses to stability evaluations of rock slopes, rock collapses, and tunnel surrounding rock in engineering.

## 1. Introduction

Brittle failure is crucial in engineering, especially in dam foundation engineering, deep underground engineering, mine openings, and slopes. Excessive displacements must be avoided, and large strains are rarely involved. In the Earth science field, brittle failure is also interesting in areas such as fault and earthquake mechanics [1,2,3]. The brittle fracture of rock can occur without significant prior deformation or warning [1]. Brittle failure is induced primarily by crack initiation and propagation [4,5]. In laboratory rock inelastic behavior studies, the topic is naturally spilt into the brittle and ductile fields. Brittleness and ductility depend largely on environmental conditions such as confining pressure, temperature, and strain rate [1,5]. The transition from brittleness to ductility is a process of reducing the brittleness degree of the rock. A method for quantifying rock brittleness based on uniaxial compressive strength (UCS) and Brazilian tensile strength (BTS) is widely used in research and practical work [6,7,8]. In recent years, many scholars have conducted in-depth research on the brittleness index. Jamshidi et al. [9] developed a new physical and mechanical parameter using the ratio of the point load index (PLI) to the porosity (n) which accurately and quickly predicts a rock’s brittleness index. Jamshidi et al. [10] conducted uniaxial compression tests and Brazilian splitting tests on layered sandstones with different lamination angles (β), and found that anisotropy significantly effects UCS, BTS, and rock brittleness. Ghobadi and Naseri [11] measured the geomechanical properties of limestone with different freeze–thaw cycles, and used a statistical parameter model established by multiple regression and an artificial neural network. A statistical parameter model based on porosity (n), P-wave velocity (Vp), and dry density (ρ_d_) better predicts the brittleness index. Research on determination methods for the brittleness index based on the full stress–strain curve can also enrich and improve the brittleness evaluation system [12,13,14]. Therefore, investigating the damage process of specimens with different brittleness degrees is critical.

The brittle failure of rock changes with the physical mechanics and structural characteristics, energy, and physical parameters [15]. The physical mechanics and structural characteristics show that, with the failure of a specimen’s internal structure, the bearing capacity decreases, the deformation rate increases, and the elastic modulus decreases. As numerous cracks appear, the porosity increases. The energy change shows that, as deformation and fractures develop, part of the rock’s internal energy is dissipated by elastic energy, acoustic energy, and electromagnetic energy. The changes in the physical parameters are reflected in the sharp increase in acoustic emission signals (such as ringing and energy counts) [16,17], increase in electromagnetic radiation intensity [18], increase of infrared radiation intensity [19], increase in permeability, decrease in wave velocity [16], and abrupt change in resistivity [20]. Thus, an efficient method of obtaining the precursor information for brittle failure is through changes in the relevant physical parameters. Combined with previous research methods, introducing new monitoring methods to study the responses of multi-physical parameters in the fracture evolution of rocks with different brittleness degrees is imperative. A rock’s damage evolution is equivalent to crack development, which causes the rock’s brittle failure. The damage process is split into stages based on cracking behaviors [21]. Rocks usually exhibit staged damage characteristics, and the staged damage behavior of rocks with different brittleness degrees has not been described. Because the damage process of rock is difficult to observe directly, acoustic emission (AE) and laser Doppler vibration (LDV) techniques provide an alternative means of reflecting this process [22,23,24].

The rock failure process includes the initiation, propagation, and fracture of internal cracks, and rocks produce AEs during this failure process. This phenomenon was discovered by Obert of the U.S. Bureau of Mines [25]; subsequently, AE technology has been applied to monitoring and predicting rock mass stability and rockburst in mine pillars. Recently, multiple studies have been conducted on the AE characteristics of the rock fracture process using AE specifications. The rock fracture damage process has been described using the ringing count, event number, and energy parameters of AE data [26,27,28]. Rising time/amplitude (RA) and average frequency (AF) are frequently used to analyze qualitative aspects of rock fracture mechanisms [29,30,31,32]. Earthquake theory *b* value analyses have also become important in rock AE characteristic research. According to studies [26,33], the *b* value changes dramatically during rock loading and decreases prior to failure. At present, the evolution characteristics of AE parameters in the staged damage process of rocks with different brittleness degrees have not been determined.

A moving object is aligned with acoustic, electric, or light waves with a particular frequency component. The item’s velocity component changes in direct proportion to the frequency of the reflected acoustic, electric, or light wave, referred to as the Doppler effect [34]. The Doppler effect can be observed everywhere. For example, a whistle’s tone changes on a highway when a fast-moving vehicle passes you because the frequency of the sound waves you hear varies. As a non-contact testing method for vibration measurements, LDV provides excellent spatial resolution while having decreased testing time, quick dynamic response, and low power consumption [35]. Adams et al. [36] proposed a structural damage and evaluation method, and concluded that structural damage could be monitored based on changes in the structural natural frequency. Salawu [37] pointed out that structural damage diagnosis methods using the natural frequency change havehigh identification accuracies at a low cost. Based on the phase-field theory and the finite element method, Thom et al. [38] analyzed the vibration response of FGM plates with cracks and stiffeners: as the plate softens, the natural frequency decreases. Phung et al. [39] used Timoshenko beam theory to study the static bending of symmetric three-layer FGM beams, and found that when the shear coefficient is sufficiently large, deflection is reduced. Phung et al. [40] analyzed the free vibration of two-layer variable thickness plates using a numerical method and found that the natural frequency decreases when the plate softens. Using the third-order shear deformation theory and finite element method, Tho et al. [41] found that finding cracks in the inner layer of the interlayer based on vibration damage monitoring is difficult. Ma et al. [24] created a technique to assess a slope rock mass’s stability using LDV and examined the quantitative correlation between the vibration characteristics and rock mass stability. Several scholars [42,43,44] have applied LDV to study unstable rock masses and achieved good results. At present, LDV monitoring technology has not yet been used in the staged damage progress of rocks with different brittleness degrees.

In this paper, combined with AE and LDV monitoring technology, uniaxial compression tests are performed on rock-like specimens with varying brittleness degrees. The variation in the AE characteristic parameters and the natural frequency of the vibration parameters during different damage stages are systematically analyzed. Multiple parameter responses are used to acquire precursor information concerning specimen instability. The relationship between the brittleness index and multiple parameters is then quantitatively analyzed, as is the relationship between the natural frequency and energy released by brittle failure.

## 2. Design of the Uniaxial Compression Test

### 2.1. Petrographic Studies

For this study, rock was taken from the Dongmiaojia landslide on the Yellow River’s right bank, approximately 2 km downstream of the Xiaolangdi reservoir dam. The rock characteristics were identified by a polarizing microscope. Figure 1 shows the microscopic image of the rock. The rock is mainly composed of fragmental and interstitial material. The fragmental material is composed of quartz, feldspar, and a small amount of mica which accounts for 50%. The interstitial material is composed of cement and matrix. The cement material is mainly calcite (45%), and the matrix is mainly a clay mineral (5%), which is reddish brown due to mixed iron oxide. The rock is silty in texture; according to the texture, structure, and mineral composition of the rock, it is named argillaceous calcareous siltstone.

### 2.2. Similarity Relationship between Sandstone and Rock-like

The argillaceous calcareous siltstone retrieved from the field was processed into 2 standard cylinders for the experiment, 1 with a 50 mm diameter and 50 mm height, and the other with a 50 mm diameter and 100 mm height. The specimens’ end faces were polished, with the non-parallelism of both ends being less than 0.05 mm. The specimens’ dimensional accuracy meets ISRM test specification requirements [45]. After measuring the specimens’ sizes and weights, the uniaxial compression, Brazil split, and direct shear tests were performed. The loading rates of these 3 experiments were 0.5 mm/min, and argillaceous calcareous siltstone’s physical and mechanical parameters (Table 1), uniaxial compression stress–strain curve (Figure 2), and failure mode (Figure 2) were obtained.

Simulating an original rock’s failure by studying synthetic rock-like specimens has obtained good results [46,47]. Fabricating rock-like specimens require calcite (i.e., coarse calcite: 20–40 mesh, fine calcite: 40–70 mesh), barite powder (200 mesh), and bentonite (400 mesh) as aggregates, plaster (120 mesh) and cement (a 425 strength grade) as binders, the ratios of aggregates to binders was 1:1, and a plaster retarder and water as additives. The water content was 11%, and the plaster retarder solution concentration was 1%. 

Figure 3 depicts the specimen processing procedure. The specimens were processed into 2 standard cylinders, 1 with a 50 mm diameter and 50 mm height, and the other with a 50 mm diameter and 100 mm height. The specimens were cured at room temperature under dry conditions, and the quality remained unchanged for 28 days. The specimens’ end faces were polished, with the non-parallelism of both ends being less than 0.05 mm. After measuring the specimens’ sizes and weights, the uniaxial compression, Brazil split and direct shear tests were performed on the rock-like specimens. The loading rates of the 3 tests were 0.5 mm/min. The rock-like specimens’ physical and mechanical parameters (Table 1), uniaxial compression stress–strain curve, and failure mode (Figure 2) were obtained. Figure 2 shows that the rock-like specimens have undergone a compaction stage, elastic stage, plastic stage, and brittle failure stage in turn, and the stress–strain curve is similar to that of the argillaceous calcareous siltstone. Figure 2 also shows that the rock-like and argillaceous calcareous siltstone have similar failure modes.

The similarity between the rock-like specimens and the argillaceous calcareous siltstone is further proved by similarity theory. Ideally, without considering the weight and size of the model, the following two relationships can be considered to determine if the rock-like specimens are similar to the original rock [48].
(1)$$a_{E}=a_{\sigma }=a_{c}$$
(2)$$a_{\varphi }=1$$

$a_{E}$ represents the ratio of the elastic modulus of the original rock to that of the rock-like material; $a_{\sigma }$ represents the ratio of the compressive strength of the original rock to that of the rock-like material; $a_{c}$ represents the ratio of cohesion of the original rock to that of the rock-like; and $a_{\varphi }$ represents the ratio of the internal friction angle of the original rock to that of the rock-like.

The similarity constants of the argillaceous calcareous siltstone and rock-like specimens are shown in Table 2.

As can be seen from Table 2, in addition to the stress similarity constants, other similarity constants better meet the requirements of Equations (1) and (2). Meeting every similarity criterion at the same time is extremely difficult. For example, the stress similarity constant in this paper is slightly larger than the criteria, but other conditions meet the similarity conditions. It can be considered that the rock-like material is similar to the argillaceous calcareous siltstone, that is, failure of the rock-like material can simulate failure of the argillaceous calcareous siltstone.

### 2.3. Test Scheme

Obtaining similar rocks with different brittleness degrees in field sampling is difficult; the research [49,50] pointed out that the brittleness of rock-like materials decreases with decreasing cement content. In this paper, three material ratios were designed by controlling the cement content to control the ratios of the aggregates to the binders (Table 3). The rock-like specimens were prepared via the preparation process shown in Figure 3.

AE monitoring and LDV measurements were performed during the uniaxial compression tests on specimens with different brittleness degrees. The specimen surfaces were treated with a coupling substance, and an AE sensor was fixed to the specimens with paper tape to keep the AE sensor from falling off during loading as well as to ensure the effective collection of signals. Prior to the test, the coupling degree of the AE sensor was checked. In addition, to reduce the impact of background noise, the AE detection threshold was set to 35 dB, and the preamplifier was amplified to 40 dB. The sampling frequency was set to 1 MHz. The LDV was placed approximately 5 m from the MTS815 rock mechanics test system. The laser was manually focused on the test specimen, and signal stability was ensured throughout the testing. In the early stage of the uniaxial compression test, manual vibrations were generated every 30 s and data were collected. The data collection frequency was accelerated in the stage in which the specimen was near failure. Figure 4 depicts the testing site.

## 3. Results

### 3.1. Physical and Mechanical Parameters

The physical and mechanical parameters of the three material ratios of the rock-like obtained from the uniaxial compression tests are shown in Table 4.

### 3.2. Division of the Fracture Evolution Stages

Using the volumetric strain variation characteristic curve for granite uniaxial compression, Martin and Chandler [21] divided the fracture process into five stages: the crack closure, elastic, crack stable growth, crack unstable growth, and post-peak stages; the basis of the division is shown in Figure 5. Using the above method, the stress–strain curves of specimens can be segmented (Figure 6). The characteristic stress values are listed in Table 5.

### 3.3. Calculation of the Brittleness Index

The brittleness index assessment approach was developed by Chen et al. [13]. This method comprehensively considers the stress increase rate between the pre-peak initiation stress value and the peak stress value, as well as the post-peak stress drop rate which has a significant impact on rock brittleness, and develops a method for calculating the brittleness index using the complete rock stress–strain relationship. The equation is as follows:(3)$$B_{i}=\frac{\frac{\sigma_{p}-\sigma_{ci}}{\sigma_{p}}}{\frac{\varepsilon_{p}-\varepsilon_{ci}}{\varepsilon_{p}}}+\frac{\frac{\sigma_{p}-\sigma_{r}}{\sigma_{p}}}{\frac{\varepsilon_{r}-\varepsilon_{p}}{\varepsilon_{p}}}$$
where $B_{i}$ is the brittleness index; $\sigma_{ci}$ is the crack initiation stress; $\varepsilon_{ci}$ is the crack initiation strain; $\sigma_{p}$ is the peak stress; $\varepsilon_{p}$ is the peak strain; $\sigma_{r}$ is the residual stress; and $\varepsilon_{r}$ is the residual strain.

The characteristic stress and corresponding strain value and brittleness index of the three groups of rock-like materials with different proportions are listed in Table 6. The relationship between the ratios of the aggregates to binders and the brittleness index is shown in Figure 7. As the ratio of the aggregates to binders increases, the brittleness index decreases.

### 3.4. Basic AE Parameters

The basic AE parameters during the different damage stages are shown in Figure 8.

Crack closure stage: A concave contour can be seen in the stress–strain curve, reflecting the fact that the rock-like specimen is compressed and the internal cracks are closed. At this time, AE has a small amplitude, which is reflected in the low ringing count rate and the low energy rate.

Elastic stage: The stress–strain curve is linear. At this point, the crystal particles and the bonds experience recoverable linear deformation, and less AE activity occurs.

Stable crack growth stage: The slope of the stress–strain curve slope decreases and veers somewhat away from its prior linear trend. With decreasing brittleness, the amplitude of the deviation from the linear trend increases, as does the AE activity, because a specimen with low brittleness can produce large plastic deformation. AE activity shows an increasing trend; this occurs because microcracks begin to form, nucleation and convergence occur, and, finally, the microcracks propagate stably [51].

Unstable crack growth stage: When loaded to the crack damage stress $\sigma_{cd}$, the crack propagation becomes unstable. Even if the stress remains constant, the crack propagation continues [21,52]. When the external load increases to the vicinity of the crack damage stress, one or more events with higher energy levels usually occur [53,54]. In this paper, the AE energy rate and ringing count rate of the specimens with different brittleness degrees increased at the crack damage stress $\sigma_{cd}$, with the specimens with brittleness indexes of 4.16 and 3.36 showing greater increases. This suggests that the already-noticeable microcracking developed more quickly for these specimens and that there was some location with an accumulation of microcracks.

Post-peak stage: The stress decreases with increasing deformation, the specimens develop a large number of microcracks, and the microcracks connect, resulting in the development of a macroscopic fissure. When the stress value drops, AE rises significantly, representing the failure of the specimen and the large amount of energy released. The ringing count rate and the energy rate released were counted when the fracturing occurred (Figure 9): a larger brittleness degree results in a larger amount of damage release energy, as well as larger increases in the ringing count rate and the energy rate.

### 3.5. RA and AF Characteristics of AE

The parameter distributions of the rising time/amplitude (RA) and the average frequency (AF) can reveal the fracture evolution mode, as shown in Figure 10. In general, AE events associated with tensile failure have lower RA and higher AF values. Meanwhile, AE events associated with shear failure have higher RA and lower AF values [29,30,31,32].

Figure 11, Figure 12 and Figure 13 depict the RA-AF parameter distribution of the specimens. The initial data distribution map, as shown in Figure 11a, Figure 12a and Figure 13a, as well as the actual AE data gathering, is too large to determine the crack type. The density of the RA-AF parameter distribution is calculated utilizing the idea of the probability density of random data in mathematics to better represent the distribution of the AE parameters and to reflect the fracture type. Figure 11b, Figure 12b and Figure 13b illustrate the results of the calculations. The red and blue areas indicate the maximum and minimum data densities, respectively.

The fractures formed during the uniaxial compression tests are tensile-shear composite cracks and are mainly tensile cracks, as shown in Figure 11b, Figure 12b and Figure 13b. A tensile fracture’s core density area (red zone) steadily decreases as the specimen’s brittleness degree decreases. Figure 13b shows that the core density area (red region) of the shear fracture region with a brittleness index of 2.25 increases compared with those of specimens with brittleness indexes of 3.36 and 4.16. The above test findings demonstrate that, when the specimen brittleness degree decreases, the tensile failure decreases, and the shear failure increases. Under certain conditions, the specimen’s brittleness increases with the increasing loading rate [55], decreases with increasing temperature [56], and decreases with the increasing confining pressure [57]. Cao et al. [58] performed uniaxial compression tests on sandstone with varying loading rates and discovered that, with an increasing loading rate, the tensile failure was enhanced and the shear failure was weakened. Miao et al. [59] conducted uniaxial compression tests on granite at different temperatures and found that, with increasing temperature, the tensile failure was weakened and the shear failure was enhanced. Yang et al. [60] performed triaxial compression tests on red sandstone with varying confining pressures and found that, with increasing confining pressure, the tensile failure was weakened and the shear failure was enhanced. The findings of the above studies are consistent with our conclusions.

### 3.6. Characteristics of the AE b Value

The *b* value is a metric that characterizes the magnitude-frequency correlations of earthquakes. The famous G-R relationship was proposed by Gutenberg and Richter [61]. The G-R relationship in the AE method is as follows:(4)$$LgN=a-b(A_{dB}/20)$$
where *A*_dB_ is the amplitude of AE in dB and *N* is the number of AE hits or events with an amplitude greater than *A*_dB_ [62].

Calculating the *b* value, with 400 acoustic emission events as a step, $\Delta A_{dB}$ = 5 dB was selected to count the rock specimens’ AE amplitude data, and the least squares method was used for the calculation. Sliding was conducted in chronological order, and the *b* value was calculated repeatedly.

Figure 14 displays the characteristics of the *b* value over the course of the uniaxial compression tests.

The changing properties with the *b* value have particular physical implications. The fraction of small events dominated by small-scale microfractures increases when the *b* value increases. A constant *b* value denotes both a consistent microfracture condition at various scales and a constant distribution of large and small AEs. A reduction in the *b* value implies that the fraction of large events has increased, as have the large-scale microfractures. The slowly changing microfracture state indicates a progressive and stable expansion process, which is reflected by a gradual shift in the *b* value across a constrained range. A sudden transition over a large range of *b* values indicates a sudden change in the microfracture state, representing a sudden increase in the instability [63].

Crack closure stage and elastic stage: The *b* value fluctuates slightly, indicating that the internal cracks in rocks develop slowly at this stage, and the proportion of AE events with different energy changes little.

Stable crack growth stage: The *b* value shows fluctuations. Due to the different crack sizes, the proportion of AE events with different energy varies greatly, and the proportion of large and small AE events changes alternately, which makes the *b* value fluctuate.

Unstable crack growth stage and post-peak stage: In the specimens with brittleness indexes of 4.16 and 3.36, the *b* value decreases continuously prior to the peak, which means that the crack propagation inside changes; this is a manifestation of crack burst propagation and can be utilized as instability precursor information of the specimen. In the specimen with a brittleness index of 2.25, the *b* value prior to the peak fluctuates, which indicates that the instability precursor information was not obvious with the decrease in the brittleness index. The minimum *b* value appears at the moment when the stress decreases after the peak value, that is, at the moment of macroscopic fracture of the specimens, the crack penetrates and releases more energy, resulting in more high-energy AE events. As shown in Figure 15, as the brittleness index increases, the minimum *b* value decreases. This is because the specimens with greater brittleness release more energy during fracture, resulting in larger AE events. This is in line with what Miao et al. [59] concluded.

### 3.7. Change in the Natural Frequency during the Uniaxial Compression Tests

An important dynamic parameter, the natural frequency, can be obtained from the LDV measurement data using the acquisition process shown in Figure 16.

A rock structure’s natural frequency is an intrinsic attribute. The natural frequency exists regardless of whether the outer world stimulates the structure. When an external excitation occurs, the structure responds according to the natural frequency of the vibration response. Ignoring the damping coefficient, the formula of the natural frequency without damping is as follows.
(5)$$f_{n}=\frac{1}{2\pi }\sqrt{\frac{k}{m}}$$
where $f_{n}$ is the undamped natural frequency; k is the stiffness; and *m* is the mass.

It can be seen from the calculation formula that the natural frequency is proportional to the stiffness and inversely proportional to the mass. In the test, the mass remained unchanged and the change rule of the natural frequency represents the change rule of the stiffness.

The natural frequency variation of the uniaxial compression test is shown in Figure 17.

Crack closure stage and elastic stage: The specimens were compacted with increasing axial stress, which increased the stiffness and the natural frequency.

Stable crack growth stage: Loaded to a position near the crack initiation stress point, microcracks began to appear, resulting in a decrease in the specimen stiffness and natural frequency. The natural frequency showed an overall upward fluctuation state. The reason for the fluctuation of the natural frequency at this stage is that, when a specimen is loaded to the crack initiation stress point, microcracks appear and the natural frequency decreases. Under axial compression, the natural frequency increases. During this stage, the data points of the natural frequency of the three specimens were linearly fitted (Figure 18a). Figure 18b depicts the link between the slope of the fitting line and the brittleness index. Specimens with a greater brittleness degree have a larger fitting line slope, and their natural frequency increases faster. Linear fitting of the rising slope of the frequency and brittleness index result in a fitting equation of y=592.05x−35.38, with a goodness of fit of 0.99.

Unstable crack growth stage: When loaded to the crack damage point, a specimen’s natural frequency decreases. The natural frequency is still in a state of fluctuation during this stage. The specimens in Figure 17a,b with larger brittleness degrees show a stable rising fluctuation trend, while the specimen in Figure 17c with a smaller brittleness degree shows a slight downward trend. The overall trend of this stage is considerably different from that of the previous stage. The specimens’ crack sizes increase obviously under the action of the axial stress, and the natural frequency shows a stable rising fluctuation trend or slightly decreasing state after axial compaction, showing that the stiffness is close to the limit value.

Post-peak stage: The natural frequency decreases when the specimen produces a stress drop. In other words, a decline in the natural frequency indicates a failing specimen. The slope of the curve of the natural frequency change during the post-peak period indicates the rate of the natural frequency decline. For the three specimens, the slopes of the natural frequency change curves corresponding to the stress drops throughout the post-peak stage were chosen for analysis (Figure 19). The three specimens have different rates of natural frequency decline. A larger degree of the brittleness of a specimen is correlated with a quicker decline in the natural frequency. The fitting equation was obtained by fitting the decreasing slope of the natural frequency at the stress drop and the brittleness index with the ExpDec1 function: $y=5.43e^{(x/0.56)}+3105.58$, with a goodness of fit of 0.99.

In Figure 20a, the natural frequency curves are compared and analyzed. Along with an increase in the brittleness index, the natural frequency displays a general rising pattern during the stable and unstable crack growth stages. The average natural frequency during the different stages is shown in Figure 20b. There is no obvious correlation between the natural frequency and the brittleness index of the specimen during the crack closure stage and the elastic stage. The average natural frequency increases as the brittleness index increases during both the stable and unstable crack growth stages. The ExpDec1 function was used to fit the average natural frequency of the stable crack growth stage and the brittleness index, such that: $y_{1}=0.15e^{(x_{1}/0.96)}+122.45$, with a goodness of fit of 0.99. Linear fitting of the average frequency during the unstable crack growth stage and the brittleness index was used, such that: $y_{2}=5.21x_{2}+115.88$, with a goodness of fit of 0.99.

## 4. Discussion

We recorded the change of the natural frequency during various tests (Figure 17). Increases in a specimen’s natural frequency corresponded to an increase in its stiffness. An increase in the stiffness indicates that the material is more resistant to deforming, that is, an increase in the force required to cause the unit deformation of the specimen. Due to the heterogeneity and discontinuity of rocks, a rock cannot be assumed to be a spring system, and the change of energy cannot be directly obtained according to the change of stiffness and deformation.

According to the variation of natural frequency obtained in this paper, the relationship between the natural frequency and the energy released during the fracture of a specimen recorded by AE can be established. The area enclosed by the natural frequency and the peak strain is taken to be a statistic, and the ratio of this area to the peak strain represents the average increment of the natural frequency per unit strain, which is expressed as $\bar{\Delta f_{a}}$. This also represents the average increment of the unit strain stiffness. During the rock loading process, an increase in the stiffness represents an increase in the elastic strain energy [64]. The area is shown in Figure 21a. The formula for calculating the $\bar{\Delta f}$ is as follows:(6)$$\bar{\Delta f_{a}}=\frac{\int_{0}^{\varepsilon_{f}}f-f_{0}\;d\varepsilon }{\varepsilon_{f}}$$
where $\bar{\Delta f_{a}}$ is the average natural frequency increment; $\varepsilon_{f}$ is the strain at peak strength; f is the natural frequency; and $f_{0}$ is the initial natural frequency.

The average natural frequency increment grows as the specimen’s brittleness index increases. The specimen releases a substantial quantity of energy at the moment of the stress drop, and the energy rate generated at the moment of failure (Figure 9) can be recorded. As shown in Figure 21b, $\bar{\Delta f_{a}}$ and the energy rates both increase with similar trends as the brittleness index increases. The ExpDec1 function was used to fit $\bar{\Delta f_{a}}$ and the brittleness index, giving a fitting equation of: $y_{2}=1.09e^{\left(x_{2}/1.37\right)}+33.35$, and with a goodness of fit of 0.99. The ExpDec1 function was used to fit the release of energy during failure and the brittleness index, such that: $y_{1}=590.05e^{\left(x_{1}/2.06\right)}+1174.52$, and with a goodness of fit of 0.99.

To ascertain the connection between the energy rate generated at the moment of failure and $\bar{\Delta f_{a}}$ (Figure 22), these characteristics were linearly fitted, such that: y=156.31x−3099.93, with a goodness of fit of 0.99.

A curve analysis of the natural frequency indicates that the brittleness index and several statistics have strong relationships. With an increasing brittleness index, several effects occur: the slope of the natural frequency during the stable crack growth stage increases; during both the stable and unstable crack growth stages, the average value of the natural frequency increases; the average natural frequency increment increases; the decreasing rate of the natural frequency at the stress drop increases; and the energy released at the moment of brittle failure of the specimen increases. These rules can enable a better identification of the instability precursor information and better predictions of the released energy of rocks with different brittleness degrees.

## 5. Conclusions

In this paper, multiple parameters were used to investigate the staged damage process of rock-like material with different brittleness degrees. The following conclusions were drawn.

1. The brittleness degree determines the fracture mode of the rock-like materials. A quantitative study of the responses of multiple parameters during the process of the staged damage of the rock-like materials according to the brittleness index was investigated.

2. With an increasing brittleness degree, the specimens show different instability precursor information during the unstable crack growth stage: the AE *b* value changes from the fluctuating state to the continuously decreasing state, and the natural frequency changes from the stable fluctuation state to the upward fluctuation state. The ringing count rate and the energy rate of specimens with different brittleness degrees increase at the crack damage stress point.

3. At the moment of stress drop caused by the brittle fracture of a specimen, the energy rate and the ringing count rate increase suddenly. The *b* value is reduced to the minimum, and the natural frequency decreases. With increasing brittleness degrees, the increase in the energy rate and the ringing count rate increases, the minimum value of the *b* value decreases, and the natural frequency reduction rate increases. These three indicators can reflect the degree of brittle failure.

4. The natural frequency is a symbolic index for studying rock-like materials with different brittleness degrees and can reflect the staged damage characteristics and predict the energy released by the brittle failure of specimens with different brittleness degrees. The natural frequency provides a reference for the monitoring of multiple parameters in rock mechanics tests.

## Figures and Tables

**Figure 1 materials-16-02334-f001:**
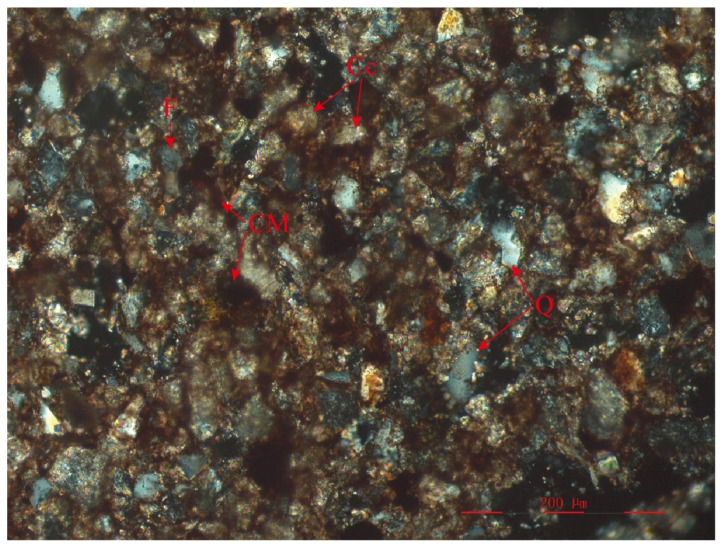
Photomicrographs of the tested specimen. Legends: Q stands for quartz, CM for clay mineral, Cc for calcite, F for feldspar.

**Figure 2 materials-16-02334-f002:**
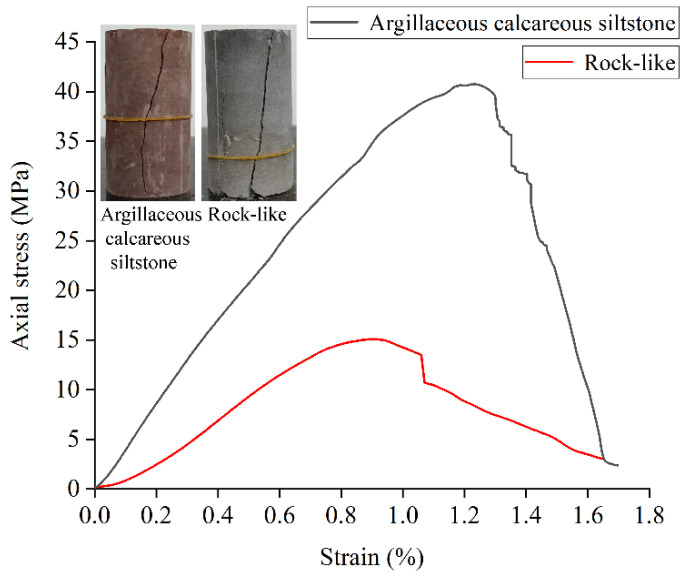
Stress–strain curve and failure mode of the argillaceous calcareous siltstone and the rock-like material.

**Figure 3 materials-16-02334-f003:**
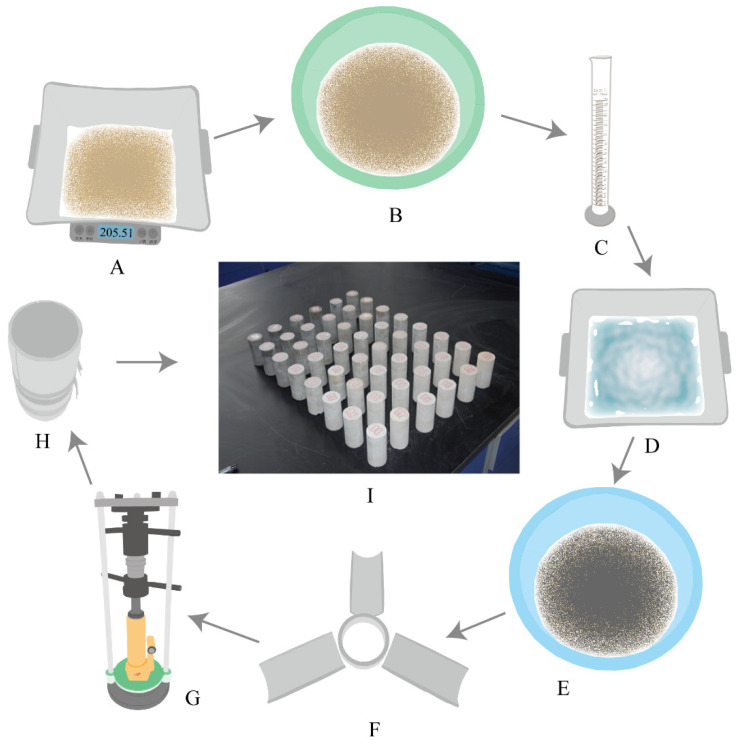
Specimen preparation process: (**A**) weighing; (**B**) dry material mixing; (**C**) water weighing; (**D**) gypsum retarder preparation; (**E**) wet material mixing; (**F**) mold preparation; (**G**) press molding with a jack; (**H**) grinding and standing; and (**I**) specimen maintenance.

**Figure 4 materials-16-02334-f004:**
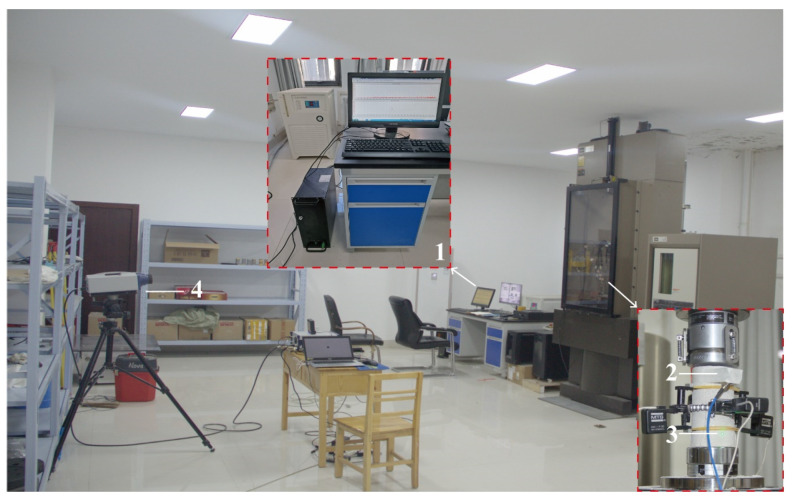
Testing site: 1—the AE system; 2—the AE sensor; 3—the LDV measurement point; and 4—the LDV.

**Figure 5 materials-16-02334-f005:**
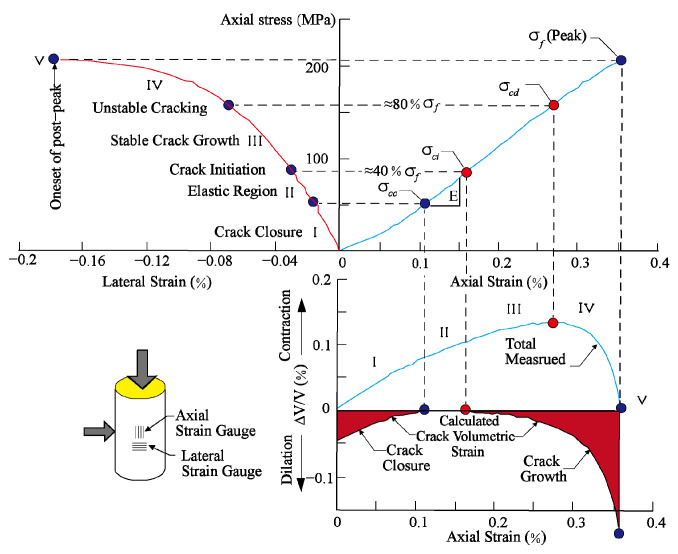
Crack volumetric strain model [21].

**Figure 6 materials-16-02334-f006:**
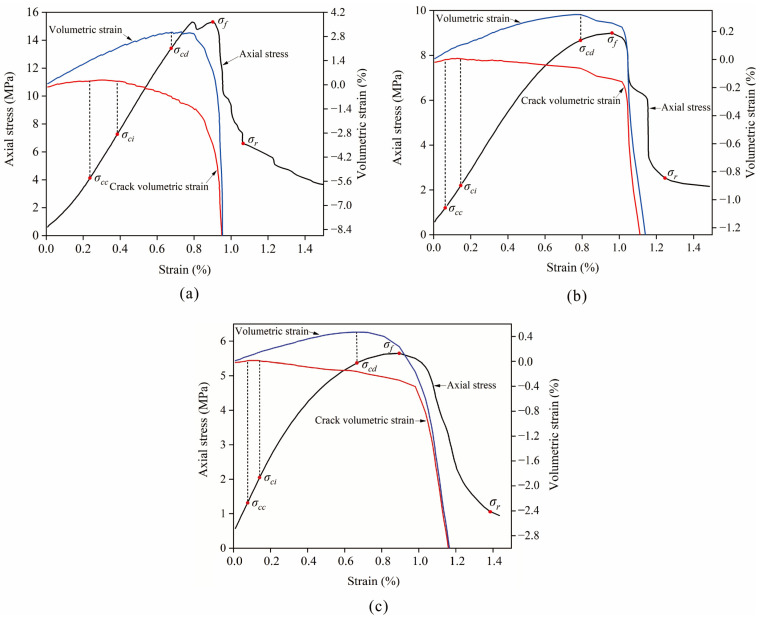
Crack volumetric strain curves, volumetric strain curves, and stress–strain curves of specimens with different ratios of aggregates to binders: (**a**) 1:1; (**b**) 4:1; and (**c**) 7:1.

**Figure 7 materials-16-02334-f007:**
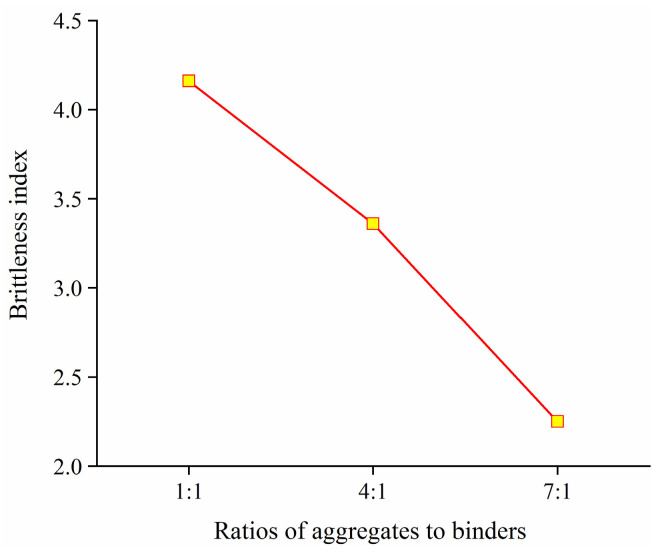
Relationship between the brittleness index ($B_{i}$) and the ratios of the aggregates to binders.

**Figure 8 materials-16-02334-f008:**
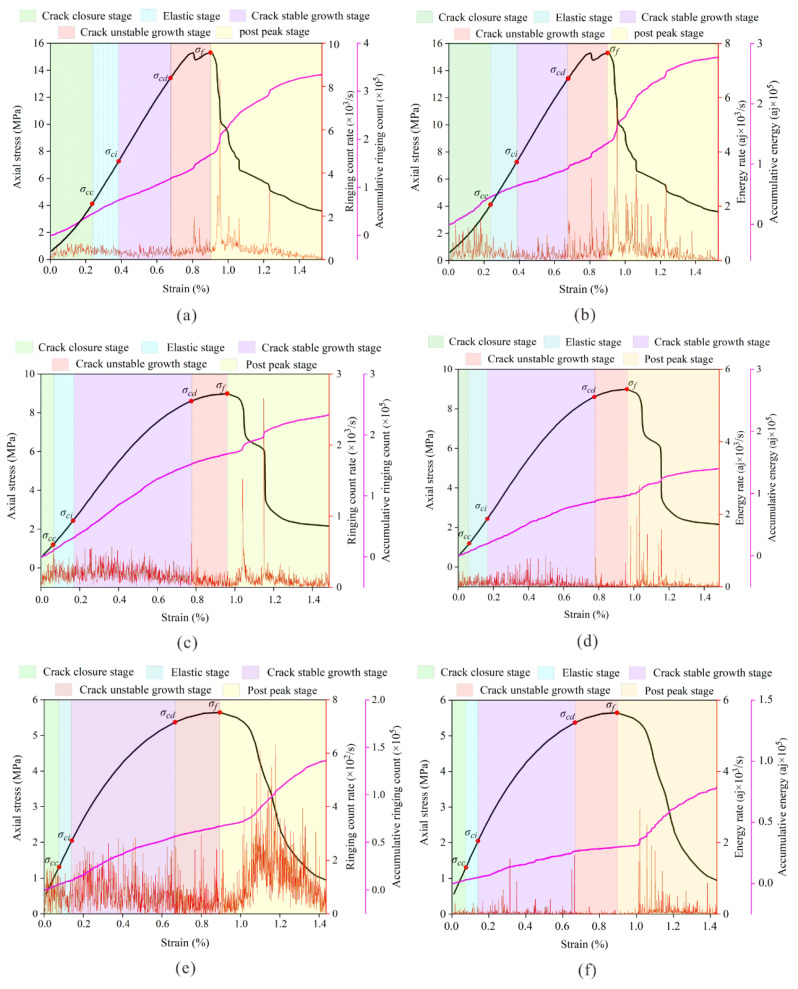
Variation of the AE parameters of the specimens with different brittleness indexes (4.16, 3.36, and 2.25) during different failure stages: (**a**,**c**,**e**) the ring count rate and accumulative ringing count and (**b**,**d**,**f**) the energy rate and accumulative energy.

**Figure 9 materials-16-02334-f009:**
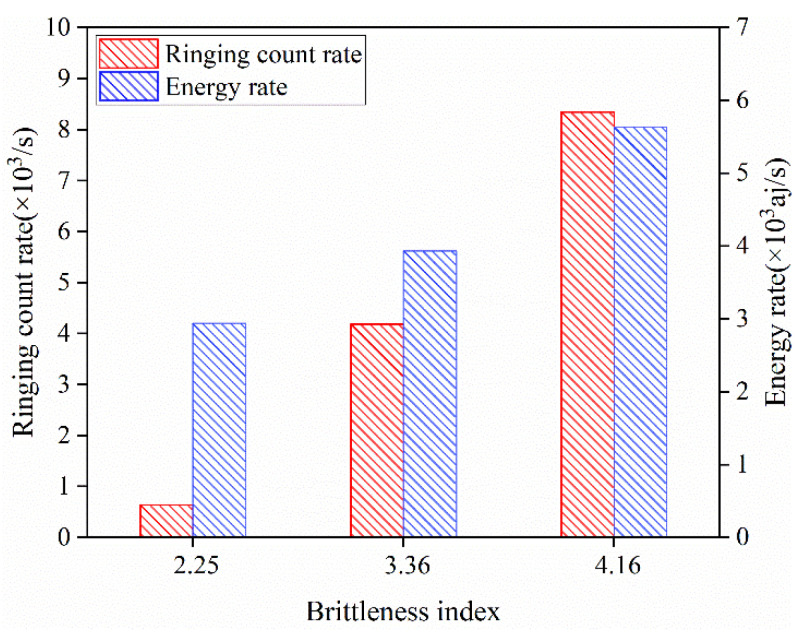
Relationship between AE released at the failure of the specimen and brittleness index.

**Figure 10 materials-16-02334-f010:**
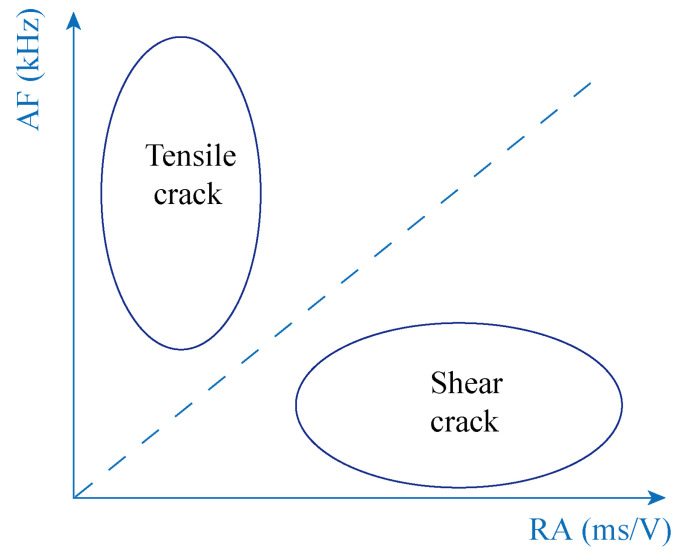
Fracture mode classification criterion.

**Figure 11 materials-16-02334-f011:**
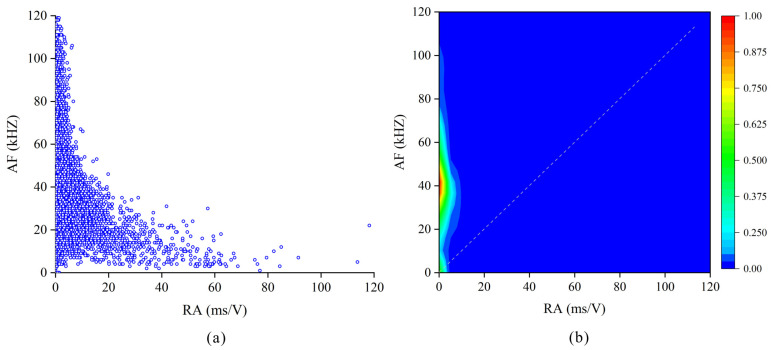
Distribution of the AE RA-AF parameters for the specimen with a brittleness index of 4.16: (**a**) scatter diagram; and (**b**) density nephogram.

**Figure 12 materials-16-02334-f012:**
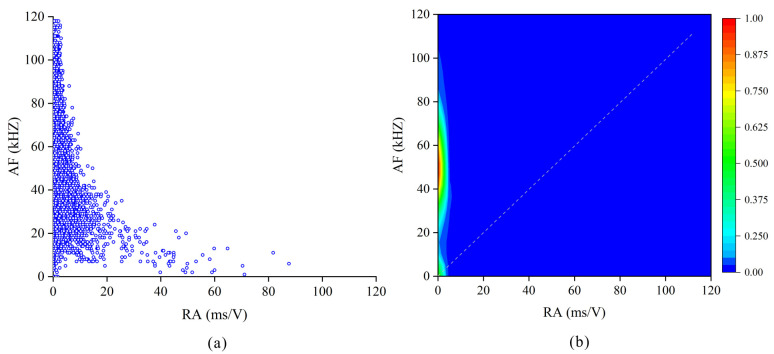
Distribution of the AE RA-AF parameters for the specimen with a brittleness index of 3.36: (**a**) scatter diagram; and (**b**) density nephogram.

**Figure 13 materials-16-02334-f013:**
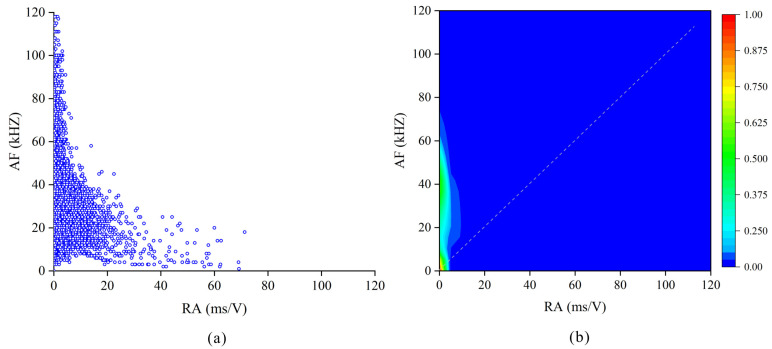
Distribution of the AE RA-AF parameters for the specimen with a brittleness index of 2.25: (**a**) scatter diagram; and (**b**) density nephogram.

**Figure 14 materials-16-02334-f014:**
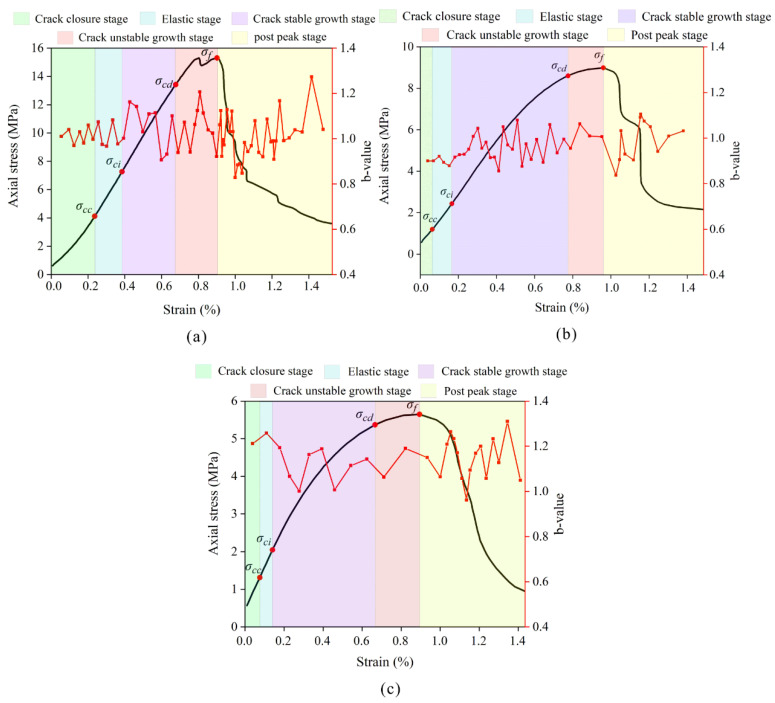
Relationship between the stress, the *b* value, and the strain for brittleness indexes of (**a**) 4.16, (**b**) 3.36, and (**c**) 2.25.

**Figure 15 materials-16-02334-f015:**
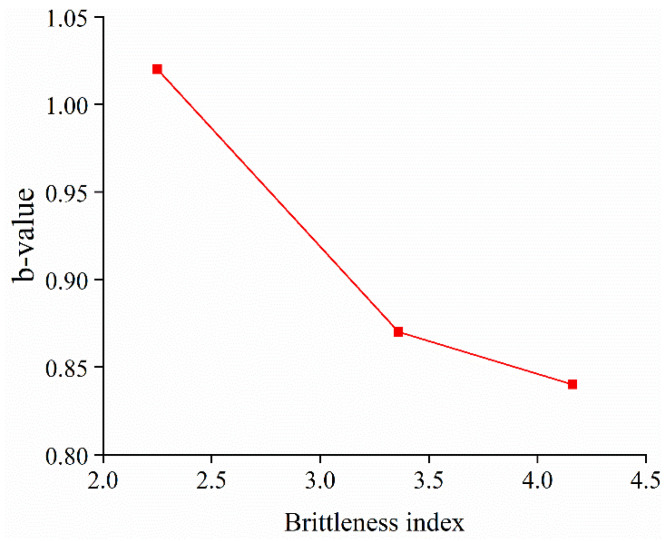
Relationship between the *b* value and the brittleness index.

**Figure 16 materials-16-02334-f016:**
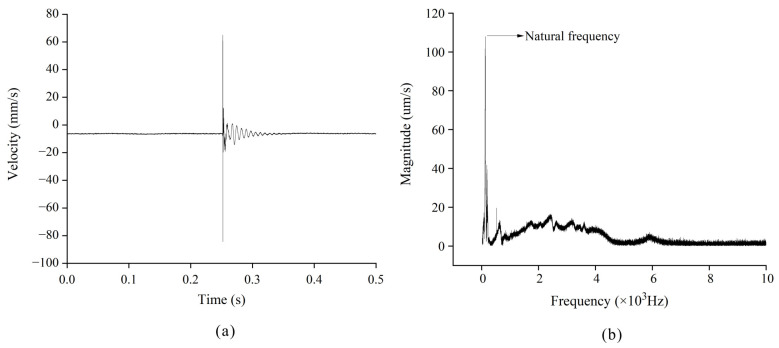
Acquisition of the natural frequency: (**a**) time-domain illustration and (**b**) frequency-domain illustration after a fast Fourier transform of the time-domain diagram.

**Figure 17 materials-16-02334-f017:**
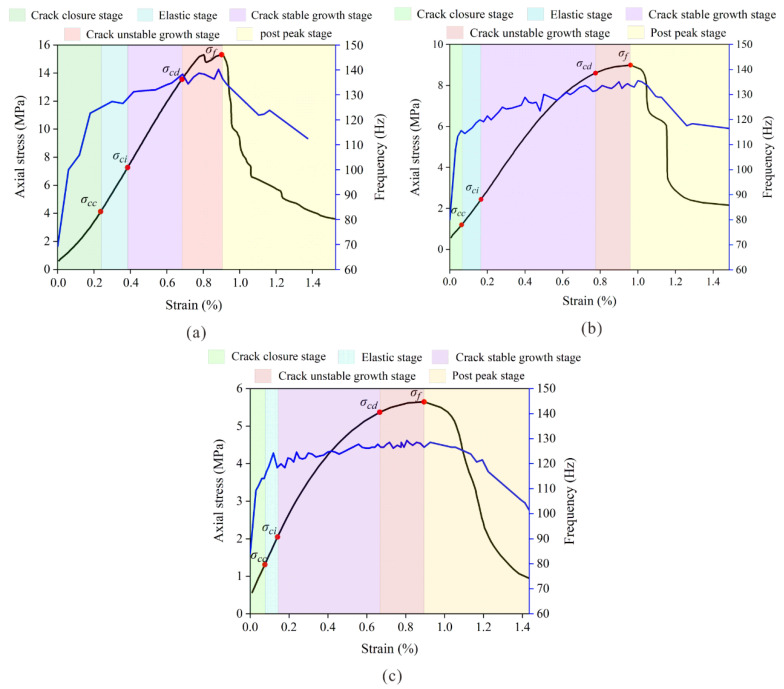
Axial stress, natural frequency, and strain curves for brittleness index of (**a**) 4.16, (**b**) 3.36, and (**c**) 2.25.

**Figure 18 materials-16-02334-f018:**
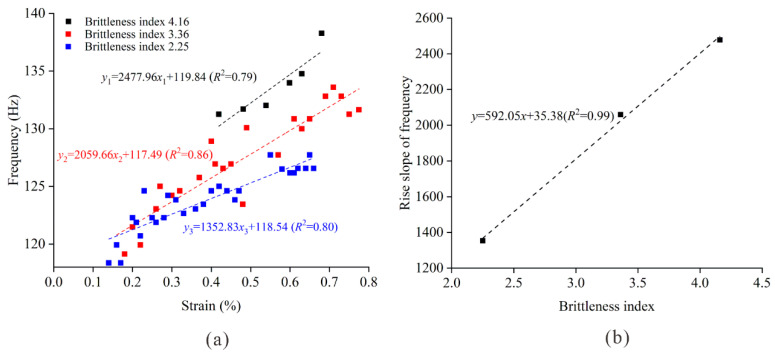
Law of natural frequency during the stable crack growth stage: (**a**) linear fitting of frequency and (**b**) linear fitting of the rise slope of the frequency and brittleness index.

**Figure 19 materials-16-02334-f019:**
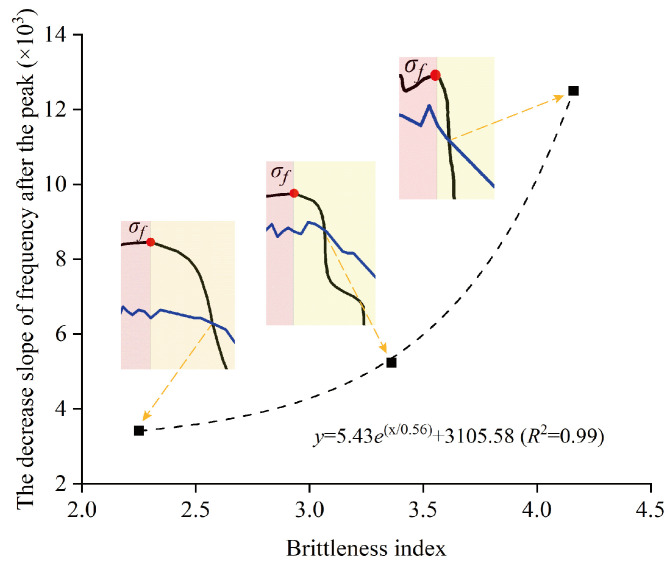
Relationship between the natural frequency decline slope at the stress drop and the brittleness index.

**Figure 20 materials-16-02334-f020:**
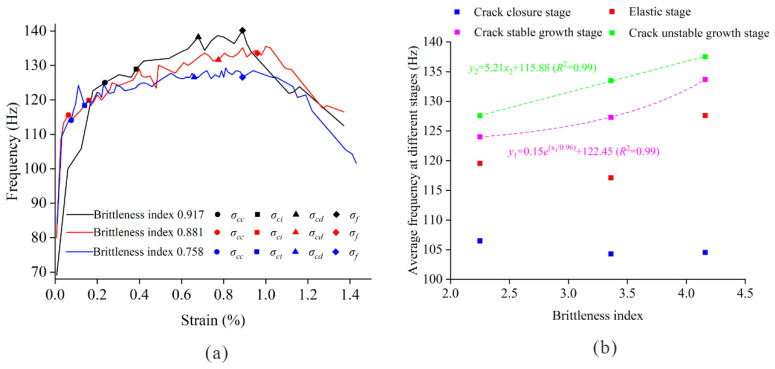
Variation analysis of the natural frequency: (**a**) natural frequency curves of specimens with different brittleness indexes and (**b**) relationship between the average natural frequency during different stages and the brittleness index.

**Figure 21 materials-16-02334-f021:**
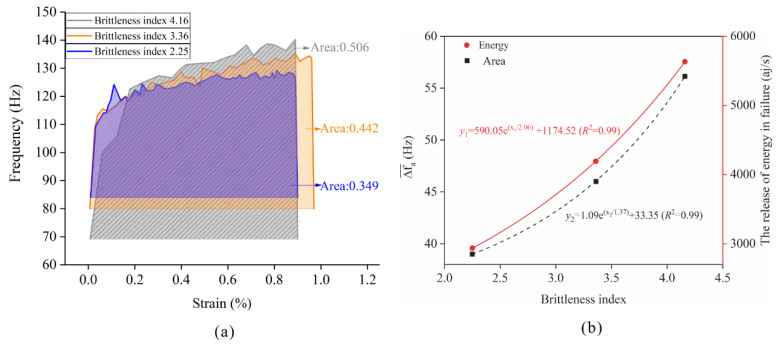
Analysis of the average natural frequency increment and energy: (**a**) area formed by the natural frequency curve and the strain at peak strength of specimens with different brittleness indexes; and (**b**) relationship between $\bar{\Delta f_{a}}$, energy and the brittleness index.

**Figure 22 materials-16-02334-f022:**
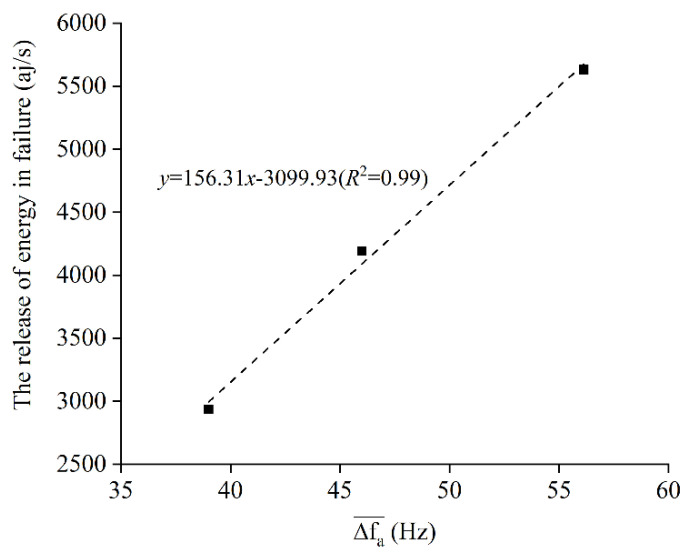
Relationship between the release of energy during failure and $\bar{\Delta f_{a}}$.

**Table 1 materials-16-02334-t001:** Physical and mechanical parameters.

Group	Density (g × cm^−3^)	Uniaxial Compressive Strength (MPa)	Elastic Modulus (MPa)	Tensile Strength (MPa)	Internal Friction Angle (°)	Cohesion (MPa)
Argillaceous calcareous siltstone	2.51	40.75	4274.60	2.03	47.71	7.57
Rock-like	1.90	15.09	2429.57	1.01	47.56	4.64

**Table 2 materials-16-02334-t002:** Similarity constants of the argillaceous calcareous siltstone to rock-like material.

$a_{\sigma }$	$a_{E}$	$a_{c}$	$a_{\varphi }$
2.70	1.76	1.63	1.00

**Table 3 materials-16-02334-t003:** Materials used to make the test specimens.

Group	Calsite (g)	Barite Powder (g)	Bentonite (g)	Cement (g)	Plaster (g)	Ratios of Aggregates to Binders
1	157.5	135	67.5	315	45	1:1
2	157.5	135	67.5	45	45	4:1
3	157.5	135	67.5	6	45	7:1

**Table 4 materials-16-02334-t004:** Physical and mechanical parameters.

Ratios of Aggregates to Binders	Density (g × cm^−3^)	Uniaxial Compressive Strength (MPa)	Elastic Modulus (MPa)	Poisson’s Ratio	Internal Friction Angle (°)	Cohesion (MPa)
1:1	1.91	15.30	2163.05	0.17	47.56	4.64
4:1	1.91	8.99	1321.58	0.21	46.99	2.95
7:1	1.90	5.65	873.53	0.25	43.27	1.71

**Table 5 materials-16-02334-t005:** Characteristic stress values of the stress–strain curve.

Ratios of Aggregates to Binders	$\sigma_{cc}$ (MPa)	$\sigma_{ci}$ (MPa)	$\sigma_{cd}$ (MPa)	$\sigma_{f}$ (MPa)	$\sigma_{r}$ (MPa)
1:1	4.12	7.26	13.42	15.30	6.60
4:1	1.20	2.44	8.6	8.99	2.52
7:1	1.31	2.04	5.37	5.65	1.07

**Table 6 materials-16-02334-t006:** Calculation of the brittleness index.

Ratios of Aggregates to Binders	$\sigma_{ci}$ (MPa)	$\varepsilon_{ci}$ (10^−3^)	$\sigma_{p}$ (MPa)	$\varepsilon_{p}$ (10^−3^)	$\sigma_{r}$ (MPa)	$\varepsilon_{r}$ (10^−3^)	$B_{i}$
1:1	7.26	3.85	15.30	9.02	6.60	10.60	4.16
4:1	2.44	1.66	8.99	9.61	2.52	12.40	3.36
7:1	2.04	1.40	5.65	8.94	1.07	13.80	2.25

## Data Availability

The data used to support the findings of this study are included within the article.

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
