# Peer review of "Study on Staged Damage Behaviors of Rock-like Materials with Different Brittleness Degrees Based on Multiple Parameters"

_materials, 2023, doi:10.3390/ma16062334_

Round 1

Author Response

Reviewer #1:

The manuscript submitted me focusing on the staged damage behaviors of rock-like materials with different

brittleness degrees based on multiple parameters. The concept of the manuscript is very interesting and definitely suited to the Materials. The quality and achieved conclusions of the manuscript are interesting. I can recommend this manuscript for publication after these comments are attended.

A: Thanks. I have revised the reviewers' valuable comments one by one. I make the following reply to your valuable comments.
1. The written English of this work is should be revised by a native speaker.

A: Thanks. The manuscript has been polished by native English speakers.
2. The literature review (introduction section) needs considerable improvement as a large amount of work has been recently done in the field such as;

Effect of anisotropy on the strength and brittleness indices of laminated sandstone, Iranian Journal of Science and Technology, Transaction A, Science, 45, 927 936.

Rock Brittleness Prediction Using Geomechanical Properties of Hamekasi Limestone: Regression and Artificial Neural Networks Analysis. J Geope 6(1):19-33.

Prediction of brittleness indices of sandstones using a novel physico-Mechanical parameter, Geotechnical and Geological Engineering, 38, 4651 4659.

A: Thanks. I have read the articles you recommended carefully, which has benefited me a lot, they enrich the introduction of the research status in this field. Therefore, I quote them. And I also cite other related articles in this field. See lines 43-57.
3.
A section titled "Petrographic studies" should be added to the manuscript in detail with microscopic thin section images.

A: Thanks. The section titled "Petrographic studies" is added in the manuscript, the rock thin is identified by microscope, and the mineral composition of rock is analyzed. See lines 141-154.

Best regards,

Tong Jiang

Reviewer 2 Report

This paper presents experimental results of staged damage behaviors of rock-like materials with different brittleness degrees based on multiple parameters. Overall, the article is quite interesting and meaningful. The results presented are quite impressive and reliable. This article can be considered for publication after considering some minor issues as follows:

1. The plagiarism test result is currently 27%, which is within the acceptable range. However, the authors should improve the manuscript so that the index is less than 25% (Details in the attached file).

2. Abstract should be shortened and potential applications of this research should be added.

3. Table 1 should be cited.

4. Do rock-like specimens manufacture to any technical standards? It should be mentioned in the manuscript.

5. The results and discussion sections are detailed and richly presented. They are reliable.

6. Some grammatical errors need to be revised.

7. Some high-quality work on cracks can be added in the paper to enrich the introduction as follows:

"Finite element modeling of the bending and vibration behavior of three-layer composite plates with a crack in the core layer"

"Finite element modeling for free vibration response of cracked stiffened FGM plates"

"Static bending analysis of symmetrical three-layer FGM beam with shear connectors under static load"

"Numerical Investigation on Static Bending and Free Vibration Responses of Two-Layer Variable Thickness Plates with Shear Connectors"

Author Response

Reviewer #2:

This paper presents experimental results of staged damage behaviors of rock-like materials with different brittleness degrees based on multiple parameters. Overall, the article is quite interesting and meaningful. The results presented are quite impressive and reliable. This article can be considered for publication after considering some minor issues as follows:

A: Thanks. I have revised the reviewers' valuable comments one by one. I make the following reply to your valuable comments.
1. The plagiarism test result is currently 27%, which is within the acceptable range. However, the authors should improve the manuscript so that the index is less than 25% (Details in the attached file).

A: Thanks. I have reduced the repetition rate of manuscript to below 25%.
2. Abstract should be shortened and potential applications of this research should be added.

A: Thanks. The Abstract has been shortened, and potential applications of this research have been added to the Abstract. See lines 8-27.
3. Table 1 should be cited.

A: Thanks. The contents of Table 1 are all obtained by the author of the manuscript through tests.

  1. Do rock-like specimens manufacture to any technical standards? It should be mentioned in the manuscript.

A: Thanks. The dimensional accuracy of specimens meets the requirements of ISRM test specification. The above content has been added in the manuscript, and with relevant references. See lines 164-165.

  1. The results and discussion sections are detailed and richly presented. They are reliable.

A: Thank you for your affirmation of the discussion and results of my paper.

  1. Some grammatical errors need to be revised.

A: Thanks. The full text has been polished with the help of native English speakers, and some grammatical errors have been corrected.

  1. Some high-quality work on cracks can be added in the paper to enrich the introduction as follows:

"Finite element modeling of the bending and vibration behavior of three-layer composite plates with a crack in the core layer."

"Finite element modeling for free vibration response of cracked stiffened FGM plates."

"Static bending analysis of symmetrical three-layer FGM beam with shear connectors under static load."

"Numerical Investigation on Static Bending and Free Vibration Responses of Two-Layer Variable Thickness Plates with Shear Connectors."

A: Thanks. I have read the articles you recommended carefully, which has benefited me a lot, they enrich the introduction of the research status in this field. Therefore, I quote them in my manuscript. See lines 114-123.

Best regards,

Tong Jiang

Round 2

Reviewer 1 Report

According to the modifications made by the authors, the paper is recommended for publication.